# Development of a Vertically Integrated Pharmacy Degree

**DOI:** 10.3390/pharmacy9040156

**Published:** 2021-09-23

**Authors:** Daniel Malone, Kirsten Galbraith, Paul J. White, Betty Exintaris, Joseph A. Nicolazzo, Tina Brock, Andreia Bruno-Tomé, Jennifer L. Short, Ian Larson

**Affiliations:** 1Pharmacy and Pharmaceutical Sciences Education, Faculty of Pharmacy and Pharmaceutical Sciences, Monash University, Parkville, VIC 3052, Australia; dan.malone@monash.edu (D.M.); kirstie.galbraith@monash.edu (K.G.); paul.white@monash.edu (P.J.W.); tina.brock@monash.edu (T.B.); andreia.bruno@monash.edu (A.B.-T.); jennifer.short@monash.edu (J.L.S.); 2Drug Discovery Biology, Monash Institute of Pharmaceutical Sciences, Monash University, Parkville, VIC 3052, Australia; betty.exintaris@monash.edu; 3Drug Delivery, Disposition and Dynamics, Monash Institute of Pharmaceutical Sciences, Monash University, Parkville, VIC 3052, Australia; joseph.nicolazzo@monash.edu

**Keywords:** curriculum development, pharmacy degree, health curriculum transformation, pedagogical and instructional principles, skills development, faculty development

## Abstract

Whilst curriculum revision is commonplace, whole degree transformation is less so. In this paper we discuss the rationale, design and implementation of a unique pharmacy program by a research-intensive faculty. The new Monash pharmacy curriculum, which had its first intake in 2017, was built using a range of key innovations that aimed to produce graduates that demonstrate key conceptual understanding and all the skills required to deliver world-best patient outcomes. The key elements of the re-design are outlined and include the process and principles developed, as well as key features such as a student-centred individualised program of development arranged around specific, authentic tasks for each skill and earlier enhanced experiential placements where students become proficient in entrustable professional activities. It is hoped the dissemination of this process, as well as the lessons learnt in the process, will be useful to others looking to transform a health curriculum.

## 1. Introduction

Incremental changes to health care curricula are common, but whole-degree transformational change is more difficult. Hubers (2020) describes such a transformational change as second order, a step above lower-level course changes [1]. The faculty of pharmacy and pharmaceutical sciences at Monash University is a successful faculty in terms of research and teaching, often ranked within the top three in the world in pharmacy and pharmacology in the QS Top Universities rankings [2]. While the Bachelor of Pharmacy degree has produced many high-quality graduates as indicated by employment and career achievements since its inception in 1965, stakeholder engagement has revealed the need to refocus the development of graduates in a number of areas. Specifically, stakeholders, including community pharmacy and hospital employers, asked for greater development of core skills to enable improved patient care. This included a focus on skills development (communication skills in particular) and more explicit linking of content to the context of patients and/or medicines. To achieve this, our leadership team decided to implement a transformational change to what and how pharmacy students learn. The purpose of this article is to describe the development of the Bachelor of Pharmacy/Master of Pharmacy degree implemented at the Australian and Malaysian campuses of Monash University with the commencing cohort of 2017. The article particularly focuses on the need for change, pedagogical and instructional principles, distinctive implementation of skills development, and quality assurance processes and timelines. We outline these to share learnings for other faculties considering holistic health curriculum transformation.

## 2. Methods

### 2.1. Planning of the New Monash University Pharmacy Degree

To enable such widespread degree changes, senior management of the faculty were engaged to support the transformation of the pharmacy degree. Resources in the form of staff time (6 full-time equivalents) and monetary investment in several initiatives (for example, the redevelopment of several offices into physically and technologically connected small group classrooms) were provided. This process commenced in the second half of 2014, with two whole-day development meetings between key educators and professional staff in the faculty. Initial discussions centred around the creation of new streams of learning to be vertically aligned throughout the degree [3] and the implementation of the recently approved Monash University degree architecture. This architecture enables graduates to obtain the Bachelor of Pharmacy (Honours) [BPharm (Hons)], Master of Pharmacy [MPharm], and completion of provisional registration requirements (internship) in a five-year period. Some courses (called units at Monash University, but hereafter referred to as courses) taught at a master’s level during the four years of the bachelor’s degree were cross-credited towards a master’s (Figure 1). Registration and entry to practice with a master’s degree was perceived as advantageous and internationally competitive for graduates who were expected to have higher level knowledge and skills compatible with a master’s level qualification. Early planning also focused on ensuring an up-to-date and forward-looking curriculum, developing specific skills and knowledge in students, and building on the active learning gains made in the preceding years.

The faculty already had in place a partial component of the MPharm in the form of the accredited Intern Training Program, which provided graduates with academic credit towards a postgraduate qualification. Throughout 2015, additional work-integrated learning courses were developed, and the new Intern Foundation Program (IFP) was offered for the first time in 2016 (see Figure 1, Year 5). Early implementation of the IFP [4] provided an opportunity to gain experience in offering the MPharm year and gave the faculty a target to aim for when developing the new BPharm (Hons).

### 2.2. Development of a Pedagogical Framework

Recognising the evolving understanding of student learning at the time, the faculty implemented a transformation of lectures into active learning classes. Starting in 2012, “flipped” classroom active learning was embedded across each year of the pharmacy program, concluding with the rollout of the flipped fourth year courses in 2016. This transformation was founded on research that clearly demonstrated superior outcomes using active learning approaches [5,6]. Staff development focused on effective active learning approaches, such as audience response systems, concept mapping and scenario-based learning [7], with a view to developing applied knowledge, critical thinking and communication skills. Student attitudes to active learning radically improved over this period, with students acknowledging the effectiveness of active learning rather than traditional teaching for their learning. This work set the stage for the complete redevelopment of the pharmacy program in which the curriculum was rebuilt, and the teaching approach further transformed [7,8,9].

During 2016, the teaching framework for the new degree was designed and refined; the new degree was ready for implementation in 2017. The traditional teaching model in the Bachelor of Pharmacy prior to 2017 (for students enrolled in full-time study) consisted of four six-credit point courses each comprising three lectures per week. The model also consisted of a varying number of practicals or workshops (ranging from about 10 h to about 30 h) for a total of 600 h of learning per semester. The variety between courses led to a chaotic timetable for staff and students. To enable a student-friendly timetable and a consistent approach for staff, and to fully integrate active learning, a “2 + 2 + 2 model” was developed for all new first-year courses. Each six-credit point course was planned to include two hours of “Discovery” (online preparatory material supporting the ‘flipped’ classes), two hours of interactive lectures (each lecture to have a minimum of three active learning activities that engaged the whole class, termed “Explore”), and two hours of workshops per week (termed “Apply”). The intention was for students to tackle content prior to class using the discovery content and formative quizzes, gaining key terminology and concepts and identifying areas that were confusing. Interactive lectures allowed students to work with peers on activities that consolidated their conceptual understanding and clarified misconceptions. Workshops were the culmination of the topic; classes, in which application of knowledge and skills development occurred, were grounded in authentic cases [10,11]. Coupled with the inherent directive for students to reflect on their skills through skills coaching sessions (see below) and on their learning after each topic, this “Discover”, “Explore”, “Apply” and “Reflect” model (termed “DEAR”) was consistently embedded as the curriculum was developed.

Course-building teams collaboratively divided the weeks of the semester between the concepts required to be taught in the course. This model permitted the creation of a weekly repeating timetable for students and staff, including the same day each week dedicated to learning discovery content (no attendance on campus required). Concomitant with the course development period, the program adopted a bring-your-own-device (BYOD) policy that required all pharmacy students to have access to a mobile computer/laptop/tablet.

### 2.3. Building the New Degree

#### 2.3.1. Content Development and Quality Assurance

To build the curriculum, specific course coordinators were selected with due consideration regarding their ability to apply the broader principles and philosophies of the Monash University Pharmacy degree. Expert knowledge of the topics being developed was an advantage. The course coordinator then recruited a team of topic developers who were experienced practitioners and/or educators in the topic, so that relevant and contemporary teaching material could be designed. Typically, the course team included external members from stakeholder groups (predominantly hospital and community pharmacists). Engaging stakeholders early and often in curriculum development has been recently recommended when considering curriculum revision [12]. The course coordinator was supported by a project manager for processes and timelines, and by assistant lecturers and education designers to translate the material into E-learning formats.

Once a course-building team was assembled, learning outcomes were refined and created for each sub-topic as the basis for developing the teaching material and activities. Once workshops and assessments were designed, it was possible to determine the pre-learning materials (“Discovery”) and interactive lecture activities required to assimilate the clinical cases (or science-oriented material). In essence, the topics were built using a backwards design approach, but with a keen eye on the learning outcomes. In terms of conditions or disease states, rather than relying on the historic topic list or attempting to include as many as possible into the curricula, a strategic approach was undertaken. Careful review of Australian health priorities [13], Indigenous health priorities [14], global health priorities [15] and the ACCP Pharmacotherapy Didactic Toolkit [16] led to consensus on 38 key conditions to be included in depth in comprehensive care courses (Figure 1) by the end of semester 1 in year 3. These conditions were called the ‘Monash-38’, and examples include asthma, hypertension and major depressive disorder. Other conditions, health matters, and populations were spiraled through the latter year comprehensive care courses in parallel with the increasing independence and ability of students to source and evaluate relevant information.

Diversity in the delivery of teaching material was also necessary to maintain engagement and motivation. As such, the pre-learning material included text, videos, links to external sites, activities, tasks and multiple-choice questions to gauge understanding. Of particular importance was the choice of tasks, as these needed to be directly linked to exercises within the subsequent interactive lectures. Given the philosophy of not introducing new material into lectures, discussing or solving the tasks within the lecture was an ideal way of checking understanding.

All health science-based degrees require a background in topics such as chemistry and biology. Our driving philosophy when teaching foundational science material was that the science should not be taught in isolation; “no content without context” was an oft-used phrase. All basic science was taught in the context of a patient or a medicine, thus ensuring interdisciplinary integration [17,18]. This often-involved co-teaching, with experts in chemistry and pharmacology working together with pharmacists. The advantage of this approach was that the science was always viewed as having relevance to either a patient, an organ, pharmacokinetics, toxicity or a dose form, and so forth. This did not mean that less science was taught in the pharmacy degree. Rather, it was taught in context and included more science from social, administrative, epidemiological, and clinical areas in order to meet the needs of twenty-first century health professionals [19].

Imperative to the whole process was the need to regularly discuss progress within the course team. Sharing of documentation through a common platform (e.g., Google Docs or SharePoint) allowed for version control when more than one person was contributing to a document. Weekly meetings were strictly adhered to in order to gauge progress and communicate the needs of the build process. This included the development of exam questions, deadlines for getting material to the assistant lecturers, deadlines for quality assurance meetings and agreement on in-semester assessments. During these meetings, four questions were asked of course coordinators (What tangible has been produced for the course since our last meeting? What bottlenecks are slowing progress? What resources are needed to advance the work? What feedback from your team do we need to address?). Quality assurance meetings were vital, as they allowed relevant staff from all teaching sites to read and critique the material. Typical concerns were about alignment to the learning outcomes, length of the material and the diversity of activities.

#### 2.3.2. Skill Development and Assessment

During development meetings in 2014 and 2015, stakeholders were asked to consider the core skills that pharmacists require to provide better patient care and function effectively in a variety of healthcare environments. Based on these discussions and a review of the literature [17,18,20,21] eight core skills were identified; see Table 1.

The acronym “POWERIT-Inq” was used when describing these core skills to staff and students. Deliberations ensued as to the level of skill development required at stages of the degree, in addition to how these skills would be assessed. An example of mapping of a particular skill (oral communication) over the first four years is shown in Table 2. This mapping was developed taking into consideration the requirements of the National Competency Standards Framework for Pharmacists in Australia [22], the Australian Qualifications Framework requirements for level eight [BPharm(Hons)] and nine [MPharm] degrees [23], a skills development framework [24] and skills mapping for the previous Monash University pharmacy degree [25]. In order to map the teaching, practising and assessment of each skill within the degree, and to provide an electronic portfolio for student reflections, a web-based application was developed: the Curriculum Outcome Mapping and e-Portfolio Tool (COMeT). The mapping component of COMeT identifies instances of learning outcome attainment and skill development within the pharmacy degree for students, staff and accreditors. Students use COMeT to search upcoming weeks of their semester for opportunities to practice and receive formative feedback on their skills. Students use this information to plan further skills development. The student portfolio in COMeT and skills development will be further described in a later section.

Assessments taken shortly after entry into a degree have been shown to assist in identifying the type of support required for students to improve their skills [26]. In order to provide a personalised approach for students in terms of their skill development and to benchmark student progress through the degree, the faculty implemented Situational Judgement Scenario (SJS) testing at the commencement of each academic year, and English language testing (see below) at the start of year 1. Collaboration with an external agency (Work Psychology Group [WPG], Derby, UK) with expertise in the development of SJSs resulted in the implementation of a validated program of formative assessment [27]. Initial discussions with stakeholders and WPG identified problem solving, teamwork, integrity and empathy as the target skills for SJS testing. Using a validated methodology, a bank of SJSs was developed and tested and then delivered as a pilot to students enrolled in the established pharmacy degree in 2016. A concordance panel comprised of practising pharmacists provided the benchmark against which students were scored. Following the pilot, a shorter validated test was introduced in 2017 and has been in place for all cohorts, including provisionally registered (intern) pharmacists, since then. Students receive feedback for each skill based on how closely their responses match the concordance panel. They can visualise their performance against their year-level cohort and the entire degree cohort, thus identifying the areas they most need to develop. Students are also provided with prompts to consider how they might develop each skill and can track their progress year-on-year [27].

Post-enrolment language assessments have been used to assess communication competencies required for students within particular degrees; however, a post-enrolment language assessment without support and scaffolding has very little impact and in fact may be detrimental, as international students and non-native English speakers can feel targeted and marginalised by the process [28,29]. Our novel approach uses a diagnostic English Language Assessment (DELA) to determine English language proficiency not simply for diagnosis, but also as a starting point for a student-centred reflective approach to skills development. Within the first week of commencing coursework, students are allowed 45 min (under examination conditions) to compose a written long-form essay response to a statement using the evidence supplied. Linguistics experts [Language Research Testing Centre, University of Melbourne] assess the test papers and provide feedback on four domains: 1, ideas and content; 2, organisation and linking; 3, grammar, vocabulary and spelling; and 4, control of academic writing. This feedback allows all commencing students to determine the areas of their written communication in which they are proficient and the areas that require attention.

Detailed DELA and SJS feedback comprise objective evidence of students’ skill levels across multiple skills. The ability for individual students to develop these skills occurs via a process of skills coaching. Students (in groups of approximately 10) meet at least three times a semester with a skills coach (an academic staff member or practitioner), before which they are required to write a reflective personalised learning plan (PLP) on one or more of the eight core skills identified above using Borton’s model of reflection [30,31]. Each PLP is linked to evidence (such as SJS results or in-class feedback from a staff member or student in a standard manner [32]) and includes a SMART plan [33] for future development. Students use the portfolio in COMeT as the repository for their PLPs; skills coaches provide structured feedback [32] prior to each skills coach meeting, and further skills-based discussions occur in the skills coaching meetings with topics being largely student driven. At the conclusion of each skills coaching cycle, students document their agreed actions against their respective PLP.

#### 2.3.3. Earlier and Enhanced Experiential Placements

One of the foundational principles in the development of the new curriculum was that experiential placements should occur as early as possible and be closely connected to the didactic curriculum. The new Student Experiential Placements (StEPs) program was significantly expanded from the previous degree, with placements commencing in the first year for the first time. The aspiration was for StEPs to better prepare students for their pre-registration year by developing their competence in performing useful and relevant tasks in the workplace. Entrustable Professional Activities (EPAs) [34,35,36] are used as the basis for these activities. EPAs are “units of professional practice or descriptors of work, defined as specific tasks or responsibilities that trainees are entrusted to perform without direct supervision once they have attained sufficient competence” [34]. The EPAs were developed with input from the profession concerning employer expectations of the competency levels of pre-registration pharmacists at the start of their internship year. EPAs were built each year as activities were taught, practiced, and assessed in the curriculum. For example, one fourth-year EPA is to “Conduct a detailed and systematic medication history which takes into account all details of patients medication use”. Preceptors provide feedback to students based on their level of independence in performing the tasks (i.e., how much they need to be supervised). The portfolio in COMeT was enhanced to support the implementation of EPAs, including a feature allowing students to invite individual preceptors to provide EPA-based feedback.

#### 2.3.4. Tools and Frameworks

Having consistent frameworks and tools to guide clinical decision making is extremely important in health professional settings [37]. This section describes the development of a patient care model, the utilisation of existing software (MyDispense) and a framework for interprofessional education [38].

##### Monash Model of Care

Recognising the need for a consistent process to teach how pharmacists approach patient care, the Monash Model of Care (MMoC) was created (Figure 2). The development process drew heavily on globally available tools and resources [38,39,40,41] and engaged practitioners and educators, resulting in a holistic model that can be used in all courses involving clinical decision making. This process is applicable to any practice setting where pharmacists provide patient-centred care and for any patient care service provided by pharmacists.

##### MyDispense

MyDispense is an online pharmacy simulation tool that was primarily developed to teach dispensing skills to students [42] but has been enhanced to allow problem solving and decision making to be practiced and assessed. MyDispense is now consistently used in workshops and compounding pharmacy laboratories for students to practice and be assessed on their clinical problem-solving skills (Table 1).

##### Interprofessional and Collaborative Care

Our curricular innovation did not occur in a vacuum. Indeed, concomitant with the pharmacy course transformation, the university was developing a comprehensive curricular framework to align key elements of all its twelve health professions training programs. The Monash Collaborative Care Curriculum Framework (CCCF) scaffolds knowledge, skills, behaviours, and attitudes with agreed learning outcomes at the novice (first year), intermediate (middle years), and entry to practice (final years) levels of training [43]. This framework has supported the design, implementation, and evaluation of several interprofessional (pharmacy and medical students) learning activities [44,45]. The CCCF has more recently been expanded to include shared goals in digital health training, a key topic in our new curriculum [46].

#### 2.3.5. Assessment

Given the focus on skill development and a more explicit linking of teaching all content in the context of patients and/or medicines as mentioned above, assessments were designed with these principles in mind. Summative assessment tasks included in-class tests, assignments, performance at workshops and practical classes, online quizzes or reflective exercises following workshops and end-of-semester examinations. A constructive alignment approach was adopted, whereby the type of summative assessment task was linked to learning outcomes and the teaching activities designed to support learning. Authentic assessment tasks were implemented throughout the degree. During the period of implementation, the university developed a secure online testing environment that is now used across the degree.

Objective Structured Clinical Examinations (OSCEs) are summative assessment tasks that are included in the first four years of the degree. However, these OSCEs have a different focus depending on the year level. First-year OSCEs are based on assessing basic communication of a simple health care issue (e.g., a primary care request for a product). Second-year OSCEs focus on counselling, medical devices, and complex history taking in addition to measuring blood pressure. Third-year OSCEs involve a greater number of possible topics than previous-year OSCEs and are designed to test critical thinking in a clinical pharmacy context, oral communication, and empathy, while fourth-year OSCEs have a variety of stations designed to test a broader range of skills. This framework of skill development is based on principles developed previously [47]. Preparation for OSCEs was enhanced using tools implemented for the established pharmacy degree [48]. For OSCEs, as well as for in-class assessments, a standard marking guide was used that included criteria and performance indicators for assessors. The guide was adapted to be specific for each particular assessment task.

## 3. Results

Extensive evaluation of the new Monash Pharmacy degree is underway and some findings have been reported [3]. There was a substantial uptake of the new Monash Pharmacy degree, with 188 students enrolling in Year 1 in 2017. Two students declined the use of their information for research purposes, and 27 students took intermission or failed courses in 2017, meaning that 159 students are reported on in Table 3.

More female students enrolled in the program than males (71% female). As detailed in Table 3, the program attracted a large number of international students, with 31% of the students being from outside Australia.

The total number of students enrolled in each of the Semester 1 and Semester 2 courses is listed in Table 4.

Students performed well in Year 1 of the transformed Monash Pharmacy degree (Table 3). For each course, the average mark was within a Distinction (70–79) or High Distinction (80–100) range, which was considered successful for the first year that the program was rolled out. The marks achieved by students did not appear variable, with the standard deviation associated with the average course mark being 15–21%.

Another measure of success is through the Monash University Student Evaluation of Teaching and Units [courses] (SETU), which is also a mechanism to receive feedback from students regarding their experience. The data obtained from SETU is used by Monash University to monitor course quality and by staff to identify what is working well and what needs to be improved in their courses. There are a number of questions asked of students; however, as an overall indicator of course satisfaction, the results of the question “Overall I was satisfied with the unit [course]” are reported in Table 5. The results, with a maximum value of 5, suggest that the students were generally satisfied with each course, with 2 courses meeting the university aspirational targets (3.80–4.69) and 3 courses requiring improvement (3.01–3.79). There was an extremely high response rate (77–90% of all students having undertaken the SETU), indicating that the data was a true representation of the student viewpoint. The feedback provided from SETU was used to improve the courses in time for the 2018 offering.

In addition to their formal evaluation of each course, students provided informal feedback during interactive lectures and workshops. Feedback was also provided through direct communication to course coordinators and teaching staff. This immediate and often unsolicited feedback was invaluable, as it provided an opportunity for course coordinators and teaching staff to make changes to ensure the optimum learning experience for students. Examples of such immediate changes included reducing the amount of discovery material within certain courses, as students reported they were struggling to keep on top of the online material in preparation for interactive lectures and workshops, reducing the content in certain workshops by approximately 30% so that students had time to meaningfully engage with the workshop material/activities, and providing more practice assessment questions in preparation for in-semester and examination assessments.

## 4. Discussion

Comprehensive curricular change is a complicated endeavour requiring shared vision, clear processes, strong communication, flexibility, and patience. In undertaking this process, we learned lessons, both reinforcing and constructive, that are worth sharing. One of the factors that supported success was the size, composition (the dean, the general manager, academic staff, professional staff), and commitment of the original eight-member steering group to work as a team [49,50]. Each person on the team represented a specific constituency (e.g., student and academic services, education technologies). In this configuration, it was possible to sit around one conference room table. Another success factor was to build the final year first (Figure 1, year 5) [51]. This clarified the target and focused early investment on developing relationships with clinical educators who could then advise and support aspects of the curricular build of earlier years.

Supporting academics through the change process was critical. Employing assistant lecturers, educational designers, and a project manager to support the design and build provided time to change instructional methods holistically. This required substantial investment but was necessary to provide the required workforce, and it had an added symbolic value in confirming to the wider faculty that this transformation was a priority. Building from local successes [8,9] was also a positive strategy. Having trusted academics coach their colleagues worked well. The approach of starting with “why” [52] enabled us to make some tough decisions about what information was respectfully retired from the previous curriculum to make space for the more time-intensive active learning methods.

Workshopping the topics included in the ACCP Toolkit [16] with several layers of practitioners reinforced that we were on the right track and informed how the topics were aggregated. Whenever possible, we adopted clear, concise frameworks (e.g., DEAR, MMoC, the Collaborative Care Curriculum) to support curricular design, instructional methods, assessment, and evaluation. Applying these models consistently provided a common language that lowered anxiety for both academics and students. We took advantage of many face-to-face (e.g., regular staff meetings, clinical educator meetings, advisory group meetings, etc.) and virtual (e.g., regular newsletters, whole-degree feedback sessions) communications methods to spread the key messages widely. We also expanded the ways that we received feedback as we went along. We held optional student/staff sessions (called SnackChats because we provided a small snack for every student that attended) to receive feedback and to provide updates several times each semester. We also encouraged workshop facilitators to provide fast feedback about their experience by responding weekly to a brief on-the-fly survey.

Despite these strategies, there were areas where we could have done even better. In some cases, we predicted human nature incorrectly and in others, we missed the mark on how long specific tasks would take. We became more efficient over time, but we struggled regularly with deadlines for launching new courses. The original steering committee was expanded to include a new degree director and a new project manager. The dean retired, as planned, from his administrative post. Although we were mostly able to manage through these changes, there were definite periods of dissonance. As with many new build activities, we were often too busy to document properly the before, during, and after phases. This proved costly when trying to share our strategies with others. An early recommendation to build the semester assessments first [53] was attempted, and while an assessment framework was often developed in the early stages of course development, it proved too culturally different to create all assessments before content. Although we invested in communications, we did not invest enough. Distinguishing between “discussions” and “decisions” proved difficult despite the creation of a project decision log. Our faculty in Australia has a sister campus in Malaysia that, historically, has taught the same content using the same methods. Although intentions were good and “handover” video calls were scheduled, we did not invest enough in multi-campus communications. Further, our aggressive timeframe meant that we were not always able to provide adequate time for the instructors in Malaysia to prepare adequately. We also underestimated the cultural differences amongst staff and students there when approaching active learning.

While full-scale evaluation of graduates from the new Monash pharmacy degree is ongoing, we have reported early predictors of student success in OSCEs and end-of-semester examinations [54]. Later research focused on the clinical pharmacists who have supervised Monash pharmacy students on experiential placements, categorising their perceptions of student knowledge, oral communication, and clinical skills [3].

The comprehensive curricular changes to why, what, and how we teach pharmacy students have been transformational and sustained. We have used some of these models (e.g., DEAR, skills coaching) to inform the redesign of our Bachelor of Pharmaceutical Sciences program and collaborative work with other health professions [44]. When faced with a need to convert quickly to online instruction in response to the COVID-19 pandemic, our faculty was able to work together effectively, in part because of the trust built through creating this new program [44,55]. We hope that sharing our systems, processes, and resources can assist others looking to undertake a transformational health curriculum rebuild.

## Figures and Tables

**Figure 1 pharmacy-09-00156-f001:**
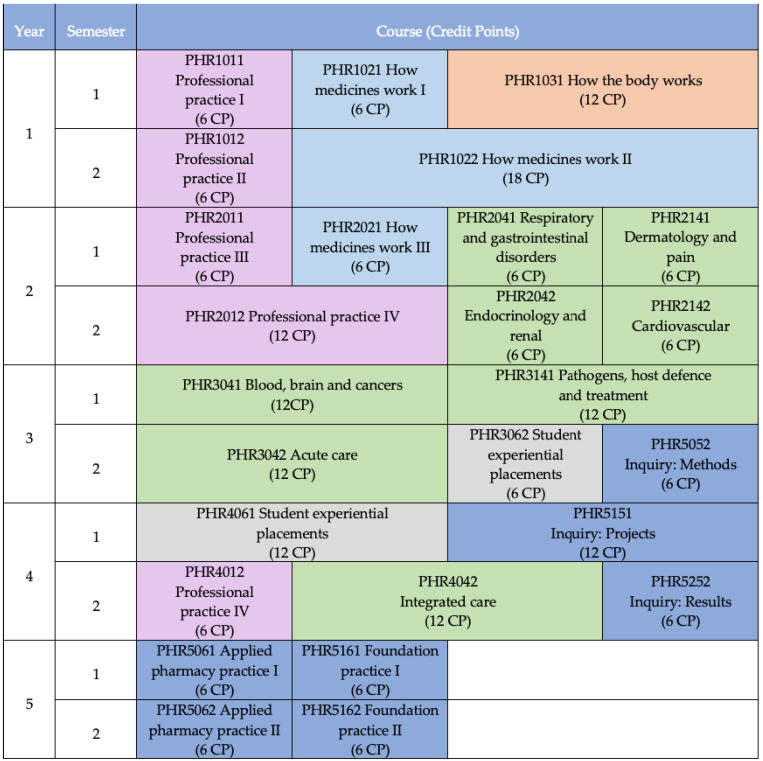
Monash University Bachelor of Pharmacy degree map. Students complete the first four years and graduate with a Bachelor of Pharmacy (Honours), and if all courses shown in year five are completed alongside the internship, students graduate at the end of year five with a Master of Pharmacy degree. Courses shaded dark blue are of master’s level. Other colors refer to different course types (pink: professional practice, light blue: how medicines work, orange: how the body works, green: comprehensive care, grey: student experiential placements).

**Figure 2 pharmacy-09-00156-f002:**
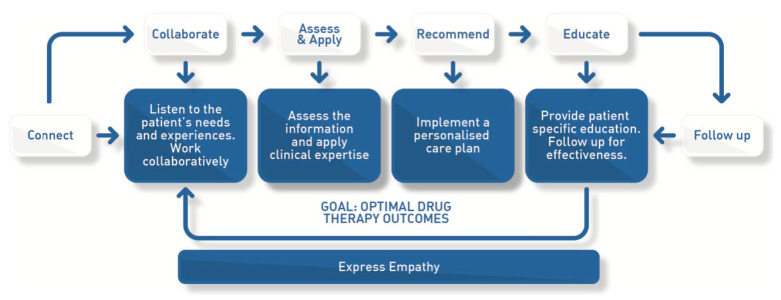
The Monash Model of Care (MMoC), a standardised process of patient care taught consistently throughout courses within the Monash Pharmacy degree.

**Table 1 pharmacy-09-00156-t001:** Eight core skills taught, practiced and assessed in the Monash University Pharmacy degree.

Skill Taught	Practiced in	Assessed via
Problem solving (and critical thinking)	Workshops	SJS, workshops, written tests and examinations, OSCEs
Oral communication	Workshops	Workshops, OSCEs
Written communication	Workshops	DELA, workshops, written tests and examinations, assignments, OSCEs
Empathy	Workshops	SJS, workshops, OSCEs
Reflective practice	Skills coaching sessions	Skills coaching sessions, assignments, OSCEs
Integrity	Workshops	SJS, written tests and examinations, OSCEs
Teamwork	Workshops	SJS, workshops, OSCEs
Inquiry	Workshops	Workshops, written tests and examinations, OSCEs

SJS: situational judgement scenarios, DELA: diagnostic English language assessment, OSCEs: objective structured clinical examinations.

**Table 2 pharmacy-09-00156-t002:** Oral communication skills mapped over the first four years of the Monash University Pharmacy degree (indicates year and semester when first introduced).

Year 1 Semester 1	Effectively orally communicates aspects of how the body works to peers Recognises the importance of good oral communication in pharmacy practice Uses clear and unambiguous oral language targeted to peers and some discipline-specific language to demonstrate understanding
Year 1 Semester 2	Diagnoses, manages, and communicates a simple heath care issue Uses discipline-specific language for a specified audience (e.g., patient, health care professional) Provides counselling for any of the first year pharmulary medicines Presents information orally in a timely, professional, and effective manner Effectively orally communicates aspects of how medicines work to peers, lay people and other health care professionals
Year 2 Semester 1	Practises history taking and decision making and communicates appropriate recommendations to lay people Adapts oral language to address lay people or health care professionals regarding disease- and medicine-related issues Practises explaining how medicines work using appropriate language to peers, lay people and teachers/lecturers Demonstrates ways of respectfully acquiring cultural information Communicates with lay people and other health care professionals in a considered and systematic way, focusing on best clinical outcomes Communicates effectively about medication management and other health care needs within a practice environment
Year 3 Semester 1	Communicates appropriate recommendations to health care professionals based on patient information Presents information orally by employing thoughtful and appropriate language to make presentation interesting and effective Presents clear and consistent information orally, demonstrating adequate consideration of audience and purpose
Year 4 Semester 1	Presents information orally by employing thoughtful and appropriate language to make presentation compelling Employs a clear organisational pattern to make content of an oral presentation cohesive
Year 4 Semester 2	Adapts oral communication to address lay people or health care professionals regarding complex disease- and medicine-related issues Practises history taking and decision making, and communicates appropriate recommendations to lay people

**Table 3 pharmacy-09-00156-t003:** Demographic data on the students enrolled in Year 1.

Student Demographics	
**Demographic**	**Percentage**
Male	29%
Female	71%
International	31%
Domestic	69%

**Table 4 pharmacy-09-00156-t004:** Details of each first-year course.

Course Name	Course Code	Credit Points	Semester	Mark (Mean ± S.D.)	No. Students
Professional practice I	PHR1011	6	1	77 ± 12	188
How medicines work I	PHR1021	6	1	71 ± 19	187
How the body works	PHR1031	12	1	73 ± 21	188
Professional practice II	PHR1012	6	2	81 ± 15	179
How medicines work II	PHR1022	18	2	71 ± 21	177

**Table 5 pharmacy-09-00156-t005:** Student Evaluation of Teaching and Unit [course] (SETU) data for Year 1.

Course Name	Code	Question: Overall, I Was Satisfied with the Course (Median/5 ± S.D.)	Total Student Responses (%)
Professional practice I	PHR1011	3.41 ± 0.98	90%
How medicines work I	PHR1021	3.87 ± 0.89	86%
How the body works	PHR1031	3.64 ± 1.04	86%
Professional practice II	PHR1012	3.95 ± 0.73	77%
How medicines work II	PHR1022	3.73 ± 0.91	79%

## Data Availability

Data available on request due to ethical restrictions.

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
