# Peer review of "Development of a Vertically Integrated Pharmacy Degree"

_pharmacy, 2021, doi:10.3390/pharmacy9040156_

Round 1

Reviewer 1 Report

I am pleased to review this wonderful manuscript.

I think this manuscript can be accepted as it is but if the authors can add some more information, this manuscript would be more valuable for international readers.

In Table 5:
Please clarify the values are the mean or median with SD or IQR.
If the authors have the values in past Monash education model, please show and compare the present with the past education model. 

Author Response

I think this manuscript can be accepted as it is but if the authors can add some more information, this manuscript would be more valuable for international readers.

In Table 5:
Please clarify the values are the mean or median with SD or IQR.

Values in table 5 are median and the standard deviation of each value has been added

If the authors have the values in past Monash education model, please show and compare the present with the past education model. 

Our intent was to illustrate methods by which we can get feedback from students. But we realise that the text leads the reader to expect a year-by-year comparison of the data with explanations. This is not what we intended. Please see the revised text:

“The feedback provided from SETU was used to improve the courses each year.”

Reviewer 2 Report

Interesting article describing the transformation of a whole pharmacy program. The content should be of interest to others working with educational development.

I have some comments:

Line 35: When was the Bachelor of Pharmacy degree first introduced at the university? High quality graduates - in what sense and how has this been evaluated?

Line 36: Why was it considered necessary to refocus the development of graduates? Please clarify.

The main changes in the new program compared with the old program? A short description of how teaching was conducted previously would increase the understanding of what changes that have been made.

Line 336: Interprofessional learning activities. Could you please expand a little about these IPL activities? Students from different programs attend?

Line 430: What improvement were made? What were the results of these improvements? What did the student evaluation show?

The quality of the tables must be improved to increase readability.

The results section is a bit scant and the results are not really discussed in the discussion part of the paper. It would have for example been interesting to compare the results to the question “Overall I was satisfied with the unit [course]” with results prior to the transformation of the program. If such a  comparison is not possible at this stage the results included should be motivated and put into context in the discussion. The discussion and conclusions are not supported by the results.

Author Response

Line 35: When was the Bachelor of Pharmacy degree first introduced at the university? High quality graduates - in what sense and how has this been evaluated?

Anecdotal stakeholder feedback on our graduates has always been extremely positive. This positive feedback has been supported by the number of internships our graduates obtained. Full-time employment success for our graduates is 98% (https://www.compared.edu.au/institution/monash-university/study-area/pharmacy/undergraduate). Please also see revised text below.

Line 36: Why was it considered necessary to refocus the development of graduates? Please clarify.

Revised text lines 34-38:

While the Bachelor of Pharmacy degree has produced many high-quality graduates, as indicated by employment success and stakeholder feedback, since its inception in 1965, stakeholder engagement revealed the need to refocus the development of graduates in a number of areas. Specifically, stakeholders, including community pharmacy and hospital employers, asked for greater development of core skills to enable improved patient care.”

New text has been added at the end of the introduction with a recent reference:

“Preceptors perceived that this transformation has improved the performance of pharmacy students [55]”

The main changes in the new program compared with the old program? A short description of how teaching was conducted previously would increase the understanding of what changes that have been made.

Please see existing lines 106-127:

As mentioned above, reference [55] describes preceptor perceptions of the impact of the transformation on our graduates.

Line 336: Interprofessional learning activities. Could you please expand a little about these IPL activities? Students from different programs attend?

References 44 and 45 contain details of these activities. We have now specified which students attended these sessions. Revised text:

“This framework has supported the design, implementation, and evaluation of several interprofessional (pharmacy and medical students) learning activities [44-45]”

Line 430: What improvement were made? What were the results of these improvements? What did the student evaluation show?

Our intent was to illustrate methods by which we can get feedback from students. But we realise that the text leads the reader to expect a year-by-year comparison of the data with explanations. This is not what we intended. Please see the revised text:

“The feedback provided from SETU was used to improve the courses each year.”

The quality of the tables must be improved to increase readability.

All tables have been redrawn

The results section is a bit scant and the results are not really discussed in the discussion part of the paper. It would have for example been interesting to compare the results to the question “Overall I was satisfied with the unit [course]” with results prior to the transformation of the program. If such a comparison is not possible at this stage the results included should be motivated and put into context in the discussion. The discussion and conclusions are not supported by the results.

Our intent was to present an example of the official student feedback on the quality of our first-year courses. We did not intend to detail a year-by-year comparison of the data with explanations.

Reviewer 3 Report

As expected from a highly regarded pharmacy faculty, the paper is well written. The development and content or the program is described. Student numbers, feedback, collaboration and resources are described.

  1. Some sections may be enhanced by examples, such as describing a task undertaken in the EPS ( section 2.3.3). 
  2. Line 413- i think the Table is 4 ( not 3)
  3. Student responses are described in Table 5.The Malaysian campus is mentioned in the Discussion. It would be interesting to compare student evaluation for each campus. Cultural differences would be interesting to explore - may be a separate paper- and would reflect on the generisability of  such  a degree structure in other countries.

Author Response

Some sections may be enhanced by examples, such as describing a task undertaken in the EPS (section 2.3.3).

Good idea! Thanks! We have added an example EPA. Please see added text:

“For example, one fourth year EPA is to “Conduct a detailed and systematic medication history which takes into account all details of patient’s medication use”.”

Line 413- i think the Table is 4 (not 3)

This has been fixed.

Student responses are described in Table 5. The Malaysian campus is mentioned in the Discussion. It would be interesting to compare student evaluation for each campus. Cultural differences would be interesting to explore - may be a separate paper- and would reflect on the generisability of  such  a degree structure in other countries.

Yes, we agree! We are drafting such a paper and can’t wait to share it once finished!

Reviewer 4 Report

This is an interesting commentary on topic of curriculum development. I believe that this text adds to the body of literature in this field, and it will be interesting to both pharmacists in practice and at universities. However, I have few suggestions:

  1. Please add new references in the introduction section, and try to emphasize the importance of your research with previous studies in this section.
  2. Line 51 - explain FTE and all the other abbreviations in the text at their first mentioning
  3. Tables look like figures and like they are of low quality. Also items in Table 2 should be named. Please make new tables standard for scientific papers in word document
  4. Please add limitation section to your manuscript

Author Response

Please add new references in the introduction section, and try to emphasize the importance of your research with previous studies in this section.

We have mentioned the reference from Hubers (2020) in the introduction, and referred to other articles in the manuscript that describe best practices in pharmacy curricular change (Ryan et al 2009) as well as smaller scale curricular amendments (Lim et al 2020). It is not the focus of the manuscript to compare to previous studies, just as others that have published about pharmacy curricular transformation have published as a commentary piece, see Roth et al 2014 (https://www.ncmedicaljournal.com/content/ncm/75/1/48.full.pdf)

Line 51 - explain FTE and all the other abbreviations in the text at their first mentioning

Done. Thank you for the reminder!

Tables look like figures and like they are of low quality.

All tables have been redrawn

 Also items in Table 2 should be named.

I assume you mean Table 1? All abbreviations used in Table 1 are given in the table caption.

Please make new tables standard for scientific papers in word document 

All tables have been redrawn

Please add limitation section to your manuscript

Limitations: At the time of writing the first cohort of students had recently completed year four of the transformed degree. A more complete measure of the success of the transformation will only be possible when sufficient graduates are employed when their learning can be assessed.

We have a paragraph in the discussion (lines 502-522) about what we could have been done better.